# Genomic Variability of SARS-CoV-2 Omicron Variant Circulating in the Russian Federation during Early December 2021 and Late January 2022

**DOI:** 10.3390/pathogens11121461

**Published:** 2022-12-02

**Authors:** Ekaterina N. Chernyaeva, Andrey A. Ayginin, Irina A. Bulusheva, Kirill S. Vinogradov, Ivan F. Stetsenko, Svetlana V. Romanova, Anastasia V. Tsypkina, Alina D. Matsvay, Yulia A. Savochkina, German A. Shipulin

**Affiliations:** Federal State Budgetary Institution, Centre for Strategic Planning and Management of Biomedical Health Risks, Federal Medical Biological Agency, 119121 Moscow, Russia

**Keywords:** SARS-CoV-2, COVID-19, Omicron, whole genome sequencing, molecular epidemiology

## Abstract

Analysis of genomic variability of pathogens associated with heightened public health concerns is an opportunity to track transmission routes of the disease and helps to develop more effective vaccines and specific diagnostic tests. We present the findings of a detailed genomic analysis of the genomic variability of the SARS-CoV-2 Omicron variant that spread in Russia between 8 December 2021 and 30 January 2022. We performed phylogenetic analysis of Omicron viral isolates collected in Moscow (*n* = 589) and downloaded from GISAID (*n* = 397), and identified that the BA.1 lineage was predominant in Russia during this period. The BA.2 lineage was also identified early in December 2021. We identified three cases of BA.1/BA.2 coinfections and one case of Delta/Omicron coinfection. A comparative genomic analysis of SARS-CoV-2 viral variants that spread in other countries allowed us to identify possible cases of transmission. We also found that some mutations that are quite rare in the Global Omicron dataset have a higher incidence rate, and identified genetic markers that could be associated with ways of Omicron transmission in Russia. We give the genomic variability of single nucleotide variations across the genome and give a characteristic of haplotype variability of Omicron strains in both Russia and around the world, and we also identify them.

## 1. Introduction

Two years ago, in March 2020, WHO declared a pandemic of COVID-19. The disease is still causing great concern because of the constant appearance of new severe acute respiratory syndrome coronavirus 2 (SARS-CoV-2) variants successively spreading across the world. The most recent new variant of SARS-CoV-2, named Omicron, was first identified in November 2021 in immunocompromised patients from Botswana and in South Africa (Johannesburg). On November 24, South Africa reported to WHO about cases caused by a new viral variant, and it was soon identified as a variant of concern (VOC) [1], classified as B.1.1.529 in the Pangolin lineage classification [2] and as Omicron in the Nextrstrain classification [3]. It is believed that there were several prerequisites for the emergence of the Omicron strain with its current features, such as the low level of immunization in Africa, the potential presence of a latent animal reservoir due to the large number of mutations, and the potential for circulation among chronically infected immunocompromised patients. [4]. The Omicron strain quickly spread to other countries, and by 6 December 2021, 40 countries had reported cases of Omicron infection. There are several versions of the Omicron origin. Until late January 2022, the Omicron lineage contained two sub-lineages: BA.1 and BA.2 [5]. Different subvariants of the omicron lineage, including variants BA.4, BA.5, and BA.2.75 began spreading around the world from January 2022, according to the GISAID data. By early summer, the BA.4 / BA.5 lineages became dominant in most countries. In Russia, BA.1 and BA.2 lineages were dominant from February to May 2022. However, since the beginning of summer, there has been an active growth in the proportion of the BA.5 lineage, reaching 98% by August 2022 [3].

The size of the SARS-CoV-2 genome is 30 kb, including a variable number (from 6 to 11) of open reading frames (ORFs) [6]. The first ORF encodes 16 non-structural proteins (NSPs), while the remaining ORFs encode additional proteins and structural proteins. The four main structural proteins are the spiked trimeric surface glycoprotein (S), the envelope protein (E), the matrix protein (M), and nucleocapsid protein (N) [7]. Omicron variants are characterized by the highest number of mutations across the genome relative to the Wuhan-Hu-1 reference genotype, compared with other genetic lineages, and contains at least 32 mutations in the spike glycoprotein mutation profile, while the Delta variant contains only 16 mutations [8,9]. Whole genome sequencing (WGS) revealed 14 mutations that were identified earlier (including three deletions and one insertion) [9], and 22 new mutations (including three deletions) in the spike gene [10].

The goal of this study was to provide molecular genetic characteristics of Omicron strains collected in Russia between 8 December 2021, and 30 January 2022, compare these strains with the global SARS-CoV-2 population, and identify some specific genomic features.

## 2. Materials and Methods

### 2.1. Sample Collection

Nasopharyngeal swabs from patients with confirmed COVID-19 infection were collected between 8 December 2021 and 30 January 2022 by clinical diagnostic laboratories in Moscow, with further transfer to a PCR laboratory at the Center for Strategic Planning (CSP) of the Federal Medical-Biological Agency (FMBA) of Russia. All samples were collected in tubes with transport media (Single-Use Virus Collection Tube, CDVCT-1, CDRICH) suitable for SARS-CoV-2 virus storage, transferred to the laboratory within 48 h of collection, and stored at −80 °C prior to RNA extraction and sequencing. Transportation and storage were compliant with the local normative documents for handling of biologically hazardous samples. The following information was saved for each sample: unique identifier, date of sample collection, and geographic region of origin. The samples obtained and sequenced in this study were collected in Moscow and the Moscow region, while the data downloaded from the GISAID database covered more Russian Federation regions (Appendix A).

### 2.2. Whole Genome Sequencing

The presence of SARS-CoV-2 RNA was confirmed using the AmpliTest SARS-CoV-2 test kit (CSP FMBA), with further identification of Omicron variants using AmpliTest SARS-CoV-2 VOC v.4 PCR assay (CSP FMBA) according to instructions. The PCR assay allows detection of three mutations associated with Omicron BA.1 sub-lineage, two mutations associated with Omicron BA.2 sub-lineage, and two mutations related to the Delta variant of SARS-CoV-2. RNA from positive samples containing Omicron strains were isolated using the Ribo-prep purification kit and reverse transcribed using the Ampliseq cDNA Synthesis for Illumina kit (Illumina; San Diego, CA, USA). The resulting cDNA was amplified using the Ampliseq for Illumina SARS-CoV-2 Research Panel (Illumina), which contains 247 amplicons in 2 pools targeting the whole SARS-CoV-2 genome. Library preparation was performed using the Ampliseq Library PLUS kit (Illumina, San Diego, CA, USA). Library quality was assessed by capillary electrophoresis using the Agilent 2100 Bioanalyzer system (Agilent; Santa Clara, CA, USA). Library concentration was measured with the Qubit 4 Fluorometer (Thermo Fisher Scientific; Waltham, MA, USA), using the Qubit dsDNA HS Assay Kit (Thermo Fisher Scientific, Waltham, MA, USA). Sequencing was carried out on the Illumina NextSeq 550 System with the NextSeq 500/550 Mid Output Kit v2.5 (300 Cycles) (Illumina, San Diego, CA, USA). The manufacturers’ recommendations were followed in all cases. Median genome coverage with sequencing reads was 99.68%; the following coverage of each nucleotide was received: median —6787; minimum—4; maximum—629,511; 10th percentile 1083; and 90th percentile—23,894.

### 2.3. Consensus Calling

Paired-end sequencing data were generated for a total of 589 samples. These sequences were then subjected to the following combination of read trimming and filtration: raw reads were trimmed with cutadapt (ver. 2.10) [11] to remove adapter and primer sequences. Low-quality ends were removed with prinseq-lite (ver. 0.20.4) [12]. Trimmed reads were mapped onto the Wuhan-Hu-1 (MN908947.3) reference genome with bwa mem [13]. SNV and short indel calling was done with GATK (ver. 4.1.9.0) [10.1101/201178v3]. SNV and indels were considered consensus if MQ values were higher than 50 and DP values were higher than 1000. Regions that were covered by fewer than 10 reads were masked as N. Four low-covered regions were masked as N (10,600–10,650, 21,800–21,850, 23,020–23,084, 29,829–29,879). Sequences with more than 4% nucleotides marked as N were removed from further analysis, resulting in 542 sequences. Consensus sequences were generated by in-house scripts.

For samples obtained from the GISAID database, filtration was conducted according to the following algorithm: sequences were aligned to the reference genome, sequences were filtered by length (at least 29,000 bp) and the number of uncertain bases (not exceeding 4%). Four regions were masked as N (10,600–10,650, 21,800–21,850, 23,020–23,084, 29,829–29,879).

### 2.4. Data Preparation and Phylogenetic Analysis

In February 2022, 756 genomes of the world subset of SARS-CoV-2 and 397 Russian genomes were downloaded from GISAID (Appendix A), and 542 Omicron complete genomes from Moscow, Russia were sequenced at FMBA. The complete genome of bat coronavirus RaTG13 (MN996532.2) was used as an out-group for phylogenetic analysis; three complete genomes of Wuhan SARS-CoV-2 were used as controls during the analysis (hCoV-19/Wuhan/WH01/2019, hCoV-19/Wuhan/Hu-1/2019 and hCoV-19/Wuhan/HB-WH5-222/2020). All genomes were aligned with the reference genome Wuhan-Hu-1/2019 (GenBank: MN908947) using NextcladeCLI v.1.11.0 [14] and a FastTree 2.1.11 [15] with the following command: -gtr -nt alignment_file > tree_file.

To discover possible routes of SARS-CoV-2 infection transmission and the origin of Omicron variants in Russia, comparative analysis of WGS data from Russia (both FMBA and GISAID dataset, *n* = 939) and the whole global genome dataset was performed using UShER software. Russian genomes were aligned with Wuhan-Hu-1/2019 (GenBank: MN908947) using MAFFT v7.490. Furthermore, using ‘faToVcf’ utility, we created a vcf file and identified 10 strains closely related to the Russian strains using the UShER tool [16], and obtained 461 subtrees. Using our in-house script, subtree files were joined and duplicates were removed. Sequences from the obtained subset were extracted from the alignment public-2022-03-02.all.msa.fa, after that, a multiple alignment of the most interesting global and all Russian genomes was created, including all necessary control genomes and the genome of the out-group. Using our script, we left only one sample out of all identical sequences, as a result, a set of 2248 samples was obtained. The multiple-alignment file was used to build a minimum spanning tree with PHYLOViZ tool and maximum likelihood tree with FastTree 2.1.11.

For phylogenetic analysis, we used two separate datasets from GISAID and UShER. It should be noted that for both datasets (from GISAID containing 1699 genomes and from UShER containing 2248 genomes) showed good segregation of BA1 and BA2 lineages. It is also should be mentioned that the GISAID database used in the 1699 set is more diverse than UShER used for 2248. Thus, in the 2248 tree, there are about 120 countries, while in GISAID there are only 17. Since there were no Russian samples in the UShER database, at the step of searching for the nearest neighbors in the database for the tree 2248, all Russian samples from the 1699 tree were added.

Number of SARS-CoV-2 genomes from different sources used in the study are shown in Table 1.

Visualization of phylogenetic trees was performed using specialized Python packages (toytree, toyplot). Data on the distribution of countries by continent was taken from open sources, and a different color was chosen for each continent (Africa, brown; Europe, blue; Oceania, green; Asia, yellow; South America, red; North America, pink). In the case of countries with ambiguous definition (at the junction of the continents), the color selection for continents and countries was made randomly and does not reflect the geographical and/or political opinion of the authors. Countries on the same continent are represented in shades of the same color chosen for that continent. Most plots for Russian samples used two colors—pink for Moscow samples and blue for GISAID samples. The lines show the bootstrap values.

Using the goeBurst algorithm, 2 trees were built in the desktop version of PHYLOViZ [17] for BA1 and BA2 subsets, based on the UShER dataset with 2248 genomes. Meta and fast files were used as input for PHYLOViZ, where bat samples and three Wuhan samples were also added.

Metadata and information aboutgenomed of SARS-CoV-2 strains isolated from patients from Moscow and uploaded to GISAID database are represented in Appendix A.

## 3. Results

### 3.1. Phylogenetic Analysis and Coinfection Identification

To investigate the genomic diversity of SARS-CoV-2 Omicron variants circulating in Russia, we performed molecular genetic screening tests for strains circulating in Moscow, using the PCR method targeting the following mutations in the S gene: A67V, del69-70, P681H, and N679K. This combination allows us to identify BA.1 and BA.2 Omicron variants; the detection of substitutions P681H, N679K allowed us to identify Omicron strains; and mutations A67V del69-70 allowed us to discriminate BA.1 and BA.2 variants [18]. Preliminary screening allowed the selection of Omicron isolates; a portion of them (*n* = 542) were randomly selected for further sequencing of the complete viral genomes. Because Moscow is the largest city in Russia and a major transportation hub for Russian citizens and visitors, we believe that analysis of viral strains circulated in Moscow provides a snapshot of prevalent SARS-CoV-2 strains in Russia. To obtain a better understanding of Omicron-strain transmission in Russia, we also combined data obtained by our research team with other Russian sequences and the Global Omicron sequence dataset from the GISAID database. All genomes were also classified with the Pangolin tool.

Genomes of the Russian Omicron strains included in the study have been collected obtained from different sources: isolated from Moscow patients with the following WGS and downloaded from GISAID. The plot presented in Figure 1 shows the sampling schedule in two Russian datasets.

Thus, we can see that both datasets intersect a lot. However, the GISAID dataset includes a higher number of earlier strains, where the Moscow dataset shows a higher number of comparatively recent strains.

Phylogenetic analysis was performed to characterize the population structure of sequenced isolates and to compare them with isolates from an external Russian dataset (GISAID) and global strains (*n* = 1695). Maximum likelihood estimation based on WGS data (Figure 2) allowed us to discriminate between two large lineages among sequenced SARS-CoV-2 Omicron samples that correspond to lineage BA.1 (*n* = 158, 29%) and lineage BA.2 (*n* = 55, 10%), 37 (7%) samples—belong to B.1.1.529 lineage, and 291 (54%) samples belong to BA.1.1 lineage according to Pangolin classification. One strain (LQ-22654) was not clustered with Omicron genomes, and three strains (LQ-23066, LQ-21013, LQ-21871) were clustered within Omicron but apart from BA.1 and BA.2 major groups. The proportion of strains from each group according to phylogenetic analysis is presented in Table 2. There is no detailed clinical information available about the patients infected with several viral variants. PCR tests showed unusual combinations of mutations for BA.1 and BA.2 lineages in the gene encoding spike protein: LQ-22654 contains P681H, N679K, A67V, del69-70, and L452R; LQ-21013, LQ-21871 and LQ-23066 contain P681H, N679K, A67sV, and 69–70 deletion.

Detailed analysis of WGS data for 542 strains allowed us to identify four cases of coinfection with various SARS-CoV-2 variants. Samples LQ-23066, LQ-21013, and LQ-21871. Based on phylogenetic analysis we first identified that these samples belong to a clade within the Omicron group, but there is no significant support to discriminate between either BA.1 or BA.2. Further genome sequence data showed that both isolates contain genetic variants belonging to both phylogenetic groups; therefore, they could be classified as a case of BA.1/BA.2 coinfection. Sample LQ-22654 was not clustered with Omicron major groups; however, all standard Omicron mutations were identified in the genome. Analysis of aligned sequencing reads with the reference showed that the sample carried several types of mutations associated with Omicron and Delta variants, as a representative of the Delta–Omicron coinfection case. Based on the analysis of both PCR and sequencing results from neighboring samples on the same PCR plate, contamination and laboratory error were ruled out:samples LQ-23066 and LQ-21871 were from different PCR plates,sample LQ-22654 did not have any Delta strains in the closest wells or in the same row of the plate,there were no other samples from the same plates that contained sequencing reads from various genomes (no other contamination/coinfection cases were identified).

To exclude the possibility that the same sample had been sequenced twice, the vcf files of LQ-23066 and LQ-21871 were compared with each other. There were five mutations identified that discriminate between these two samples, allowing us to conclude that the samples were sufficiently different from each other.

Analysis of sequencing reads in the S gene region allowed us to show the ratio of reads that could be classified as different SARS-CoV-2 variants. Figure 3 demonstrates the proportion of reads in different genome coordinates that belong to Delta (orange) and Omicron (blue) variants identified in the sample LQ-22654, and Figure 4 shows the proportion of reads in different coordinates of the S gene that could be classified as BA.1 and BA.2 variants, identified in samples LQ-23066 (A), LQ-21013 (B), and LQ-21871 (C).

To identify unusual combinations of mutations, we performed annotation of well-known mutations characterized as Delta SARS-CoV-2 variants within studied genomes of Omicron SARS-CoV-2 [19]. We identified 44 genomes out of 1699 clustered with omicron strains based on phylogenetic analysis and classified by the Pangolin tool that contained at least one mutation characterized as a Delta variant (Table 3, Appendix A).

Delta-specific substitutions were found in the whole GISAID dataset containing 2.9 million Omicron genomes. Thus, N:R203M was identified in 1642, S:L452R only in 7243, M:I82T only in 3771, a combination of M:I82T and S:L452R in 1173 genomes, and a combination of M:I82T, S:L452R and ORF1a:P2046L in 105 genomes. List of Omicron strains with identified Delta-specific substitution in genome is trpresented in Appendix A.

### 3.2. Investigation of the SARS-CoV-2 Omicron Variant Origin in Russia

To study the origin of the SARS-CoV-2 Omicron variant in Russia, we reconstructed the maximum likelihood phylogeny of SARS-CoV-2 Russian strains with closely related non-Russian strains from a global dataset. To select genomes that are closely related to Russian strains included in our study, we used a tool called UShER [16] that allowed us to place Russian SARS-CoV-2 samples on a reference phylogeny that contains all previously sequenced strains across the world. We picked up world genomes identified as the ten most closely related to each studied Russian SARS-CoV-2 isolate. Corresponding fasts files were extracted from UShER multi-fasta file dated March 3 (public-2022-03-02.all.msa.fa. xz). Sequence duplicates were removed using a custom script, and the final fasta file used for further phylogenetic analysis included 2248 SARS-CoV-2, 939 of which were from the Russian dataset (Moscow and GISAID) and 1306 from other countries (Figure 5). The data obtained were used for phylogenetic analysis with FastTree software to identify phylogenetic clusters with Russian and foreign isolates that had significant bootstrap support. In addition, we used the PHYLOViZ tool [17] to identify viral transmission pathways, using both genomic data and additional data with dates of viral strain identification. We built goeBURST full MST trees [20,21] separately for BA.1 and BA.2 substrains with genomes from two sources: UShER (2248 strains) and GISAID (1699 strains) (Figure 5 and Figure 6). For both BA.1 and BA.2 lineages, a minimum spanning tree was built using the UShER dataset and selected from the most closely related strains. The number of BA.2 representatives is very small in comparison with BA.1 (158 versus 1545 for the 1699 recruitment, and 109 versus 2135 for the 2248 recruitment, respectively). In both trees, clustering of Russian samples from Moscow and GISAID is observed, which was not as obvious in maximum likelihood trees.

For both BA.1 and BA.2 lineages, a minimum spanning tree was built using two datasets: one based on the GISAID global subsample and the other an UShER dataset selected from the most closely related strains.

The minimum spanning tree built with the PHYLOViZ tool shows that for BA.1 sub-lineage strains, there are two main clusters related to Russian Omicron samples (blue and pink circles). The pink-circles cluster is mainly associated with the Moscow dataset, and the blue-circles cluster with the GISAID dataset, mainly representing samples from St. Petersburg (about 50% of the dataset) and other regions. A majority of Moscow strains are related to strains sequenced in Great Britain (England, Scotland, and Wales), the USA, and Switzerland. Another group of Russian strains (GISAID) were more closely related to American strains, but connections with Great Britain and Switzerland were also identified. The analysis of possible transmission ways of the Russian viral strains (from Moscow and GISAID), showed that Russian viral genomes formed two distinct clusters which are related to strains from England and Scotland.

Phylogenetic analysis of the Russian BA.2 genetic lineage also shows that there are two genetic clusters with a bootstrap support of 0.87 (Figure 7). One of them is represented in only Russian samples from Moscow, another one is represented in the Russian isolates from both sources: Moscow and GISAID, and also includes foreign isolates from Great Britain. One of the sub-clusters includes eight Russian genomes (GISAID dataset) closely related with viral strains identified in England and Scotland. There is also one sub-cluster with several Russian strains (Moscow dataset), but only one strain from England (Figure 7). BA.1 cluster was represented in a majority of studied samples.

A more detailed phylogenetic tree built based on SARS-CoV-2 genomes for Russian strains and closely related strains from UShER is presented in the Appendix A.

In our dataset, there were no samples from a single outbreak. There were several phylogenetic clusters identified; some of them were related to isolates sequenced in other regions of the world, but some of the clusters were represented by the Russian isolates. We might draw a conclusion that the major identified transmission pathways were associated with the USA and Great Britain. However, Omicron strains from other European countries were also related to the studied groups of Russian Omicron samples.

### 3.3. Genomic Variation across Russian Omicron Strains

To study genomic variability across Russian Omicron BA.1 variants of SARS-CoV-2 variants, we performed multiple alignment with a global subsampled Omicron dataset from GISAID (lineages BA.1, BA.1.1 and B.1.1.529), including 529 genomes by 10.02.2022. To identify unique genetic features of Russian Omicron BA.1 strains, we compared frequencies of genomic variants identified in the global dataset with the Russian datasets; all identified mutations and their frequencies are listed in Appendix A.

A majority of rare mutations identified in SARS-CoV-2 BA.1 variants in various countries are also rare in Russian viral genomes. However, comparative analysis of rare mutations in the global dataset allowed us to identify a subset of strains that contain A28877T and G28878C substitutions, which encode the synonymous amino acid substitution S202S in the N protein. In the Russian Federation, its frequency is 30.3%, while in the global dataset it does not exceed 2%. Phylogenetic analysis with the UshER dataset showed that there were several un-clustered samples, but the majority of the Russian A28877T + G28878C strain genomes clustered together in two clusters: a major cluster with strains from the USA, Scotland, England, Spain and Switzerland; and a minor cluster with other Russian strains lacking the mentioned mutations (Figure 8). Therefore, we can suppose that there were several possible introductions of the A28877T + G28878C that provided considerably high prevalence in Russia, and that they might be independent. The phylogenetic tree is presented in the Appendix A.

Samples with the double mutation A28877T + G28878 were marked with a separate color (violet #8100EB) on the phylogenetic tree, Appendix A. Among Russian samples, this double mutation occurs both in the Moscow data (146 samples with the mutation vs. 395 without it) and in the GISAID data (172 vs. 217, respectively). The majority of A28877T + G28878C Omicron strains are clustered together in the large cluster; however, another smaller group is clustered close to the group of Moscow strains without A28877T + G28878C substitutions. This may be associated with the simultaneous occurrence of these mutations in multiple branches, or the origin from a common ancestor long before the division into different groups.

A similar comparative analysis was performed for SARS-CoV-2 Omicron BA.2 strains to find differences in the mutation frequencies of strains isolated in Russia and other countries. Comparative analysis of rare SARS-CoV-2 BA.2 genomic variations reveals that a rare variant (2.7%) with mutation C25416T in gene ORF3a (synonymous substitution F8F), identified in Lithuania and Singapore, became one of the most prevalent within the BA.2 cluster in the Russian dataset from Moscow (35.2%), but not so prevalent among other Russian samples (3.7%). The whole list of mutations and their frequencies, identified in different datasets (Russia FMBA, Russia GISAID, and World GISAID), is presented in the Appendix A.

Analysis of nucleic acid substitution frequency in Omicron BA.1 (Figure 8A.) and BA.2 (Figure 8B) strains aligned to the reference genome (Wuhan-Hu-1) allowed us to identify variable and conservative regions of the genome.

According to the nucleotide variability graph, the highest variability of both lineages is observed in the region of structural genes, including S, E, M, and N genes (marked with green).

Nucleotide variations A28877T and G28878G, leading to synonymous substitution S202S in the N gene, were shown to be variable in Russian GISAID and FMBA datasets, and to have low variability in the global dataset (Figure 8A). A nucleotide substitution in ORF1ab C2470T, leading to synonymous substitution in BA.1 strains, was detected among all studied datasets and varies from 22.4% in Russian GISAID to 44.4% in the Global GISAID (Figure 8A).

Viral genomic variable regions should be considered in the development of new effective vaccines based on structural surface proteins or genes.

We identified a list of variable haplotypes in the BA.1 and BA.2 genomes in both GISAID and Moscow datasets, based on a mutation frequency of 3–80% (Figure 9). The haplotype patterns in global and Russian datasets look similar, but the frequency of linked mutations is different in these two groups. For instance, the frequency of the common set of SNPs, causing amino acid substitutions, is higher in the BA.1 global genomes than in Russian viruses belonging to this genetic line. We compared genome haplotypes of SARS-CoV-2 BA.1 and BA.2 lineages from Russia with BA.1 and BA.2 from the world dataset (Appendix A). In addition, our analysis shows that haplotype variability is higher within the Russian BA.1 and BA.2 populations than in the global dataset (114 vs 86 haplotypes in the Russian and global datasets, respectively).

We performed analysis of SARS-CoV-2 Omicron variant genomic features in studied datasets related to the formation of human neutralizing antibodies. Analysis was performed using the Immune Epitope Database (IEDB, https://www.iedb.org/, accessed on 1 August 2022) [22] which allowed us to identify a list of the following types of epitopes: linear peptides and discontinuous amino acids stimulating the formation of human SARS-CoV-2 neutralizing antibodies, in order to understand immune escape among individuals vaccinated against COVID-19. Based on genomic data, we identified that Omicron strains contained only 82 epitopes out of 311 discontinuous epitopes mentioned in IEDB, where BA.1 strains included in our study had mutations in 228 epitopes and BA.2 strains had mutations in 196 epitopes (Table 4). This fact may have contributed to the high level of Omicron incidence because previously infected or vaccinated individuals were not protected from the Omicron infection with humoral immunity. Information about of Neutralizing antibody epitopes identification in SARS-CoV-2 Omicron lineages is represented in Appendix A.

## 4. Discussion

The study of genomic variability of circulating SARS-CoV-2 strains plays an important role in understanding viral evolution and might be used in the development of diagnostic and therapeutic methods as well as vaccine development. Our study gives a description of the genomic snapshot of Omicron lineages that circulated in Russia from December 2021 to the end of January 2022.

In our study, we gave a phylogenetic structure of the Omicron population in Russia at the end of 2021 and beginning of 2022. A majority of viral strains belong to the BA.1 lineage and its numerous sub-lineages, due to its earlier introduction in Russia. In our dataset, we identified one case of Delta–Omicron coinfection and three cases of coinfection with two Omicron lineages (BA.1 and BA.2). Delta–Omicron coinfection is a rare event that has been previously described [23], but we could not find evidence with strains that belong to the BA.1 and BA.2 lineages in other studies. Unfortunately, we did not have access to clinical data to understand if there are any risk factors that increase the probability of coinfection. This data doesn’t allow us to identify any statistical patterns, as there were only a few coinfection cases identified.

Analysis of mutations that characterize SARS-CoV-2 Delta variants allowed us to identify 45 Omicron strains that contained at least one mutation associated with Delta or other VOC lineages (2.6% of studied samples from Russia and the global GISAID subsample). Analysis of the whole GISAID dataset also shows that about 0.5% of Omicron strains contain at least one mutation associated with the Delta variant. The appearance of substitutions that are not typical for the Omicron genome could be the consequences of different scenarios:convergent evolution of different viral lineages and the appearance of mutations that provide advantages from the point of view of evolution;coinfection cases, where a set of nontypical major viral variant mutations were introduced into the consensus sequence due to specificsequencing procedures or assembly pipeline features;recombination events that can lead to appearance of nontypical combination of mutations in the genome;contamination causing unusual combinations of substitutions.

Interestingly, one of these mutations has already been described in Omicron strains. A recent publication showed that Omicron-L452R has enhanced fusogenicity and strengthened infectivity, suggesting that the Omicron-L452R variant is a very risky variant [24] because of the L452R location in the receptor-binding domain. In our study, this mutation was identified alone or together with M:I82T in 25 samples from Russia (GISAID dataset). One sample from the Russian GISAID dataset carried three substitutions S:L452R, M:I82T, and ORF1a:P2046L. Mutations M: I82T and ORF1a: P2046L have been described in the Delta–Omicron recombinant strain [25] but no other Delta-specific or recombinant-specific substitutions were found in our dataset of studied genomes. Substitutions N:R203M and M:I82T in Omicron strains have not been described in the literature before. The role of substitution N:R203M in Delta strains could be related to enhanced viral replication in lung epithelial cells [26]. We suggest that identified Delta-specific mutations in Omicron viral genomes need to be investigated in more detail. For this, there should be access to raw sequencing data (fastq or bam files) to see the real proportion of reads with mutations of interest.

Study of genomic variations of viral strains spread in different countries might be useful for further molecular genetic studies related to vaccine development and specific diagnostic tests. Comparative analysis of Omicron strains from Russia with the global dataset allowed us to identify a higher prevalence of substitutions A28877T and G28878C in Russia compared with other countries (30.3% in Russia vs less than 2% in the world). These two mutations encode the synonymous amino acid substitution S202S in the N protein. Besides the analysis of point mutations, we identified haplotypes with different frequencies in the Russian and global datasets (Appendix A). Only 82 of 311 discontinuous epitopes and 5 of 8 linear epitopes were found in Omicron strains, according to an analysis of viral neutralizing antibody binding sites from the IEDB. It has been previously shown that SARS-CoV-2 Omicron lineages have high transmission rates because of two factors: mutations that increase transmissivity due to tighter binding to human ACE2, and mutations that drive humoral immune escape [27]. Our in silico analysis corresponds with experiment data demonstrating that Omicron evades binding and neutralization by most therapeutic SARS-CoV-2 monoclonal antibodies. It was shown that 85% of known human RBD-targeted mAbs fail to bind Omicron [27]. It was previously shown that amino acid substitutions in spike protein reduces the neutralizing activity of several groups of antibodies against SARS-CoV-2 [28]; therefore, Omicron evades antibodies from previous infection and vaccination. It was also shown that this viral variant can use an alternative cell entry pathway, which improves its ability to infect cells in the upper respiratory tract [29], reinfect previously infected people [30], and shows fewer clinically severe clinical symptoms compared with Delta [31]. A combination of these factors resulted in the rapid expansion of Omicron, as the variant caused the largest number of confirmed SARS-CoV-2 infections in the history of the COVID-19 pandemic.

Phylogenetic and minimum spanning tree analysis showed that most of the Russian Omicron strains included in our study were related to the USA and Great Britain strains. However, Russian Omicron strains associated with other European countries were also revealed. This fact could be explained by the analysis of publicly available SARS-CoV-2 genomes that are mainly represented by data from the USA and Great Britain, whereby the probability of finding relations with these countries is higher. The limitation of this study was related to the lack of availability of GISAID data in the UShER database, and the probability of finding closely related strains using the full GISAID dataset. UShER is a very useful instrument that allows ultra-fast placement of the genome on the existing phylogenetic tree. However, it uses a limited number of SARS-CoV-2 genomes from the GISAID database (with information about high coverage), but mainly draws from public sequence databases (NCBI Virus/GenBank, COG-UK and the China National Center for Bioinformation), which means that many genomes submitted to GISAID could not be used in our phylogenetic analysis, because phylogenetic analysis would require large computing resources.

## 5. Conclusions

Our study gives a molecular genetic snapshot of the SARS-CoV-2 epidemic in Russia at the end of 2021 and beginning of 2022. We studied the genomic variability and phylogenetic structure of Omicron population isolates obtained from Moscow (Russia) and downloaded from the GISAID database. Comparative analysis of the Russian SARS-CoV-2 genomic variants with variants spread in other countries allowed us to identify possible pathways of transmission. Interesting genomic features of Omicron strains that make the Russian viral population different from the global population were identified and might be related to the development of immune response and vaccine effectiveness. Our in silico analysis showed that Omicron might evade binding and neutralization by most therapeutic SARS-CoV-2 monoclonal antibodies. Molecular epidemiological studies of SARS-CoV-2 variants specific for different geographical regions might be useful for vaccine development and specific diagnostic tests.

## Figures and Tables

**Figure 1 pathogens-11-01461-f001:**
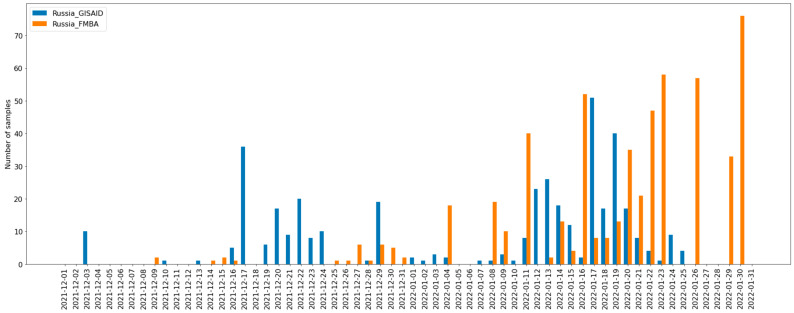
The schedule of SARS-CoV-2 Omicron samples collection in Russia, representing two datasets: GISAID (blue) and Moscow (orange).

**Figure 2 pathogens-11-01461-f002:**
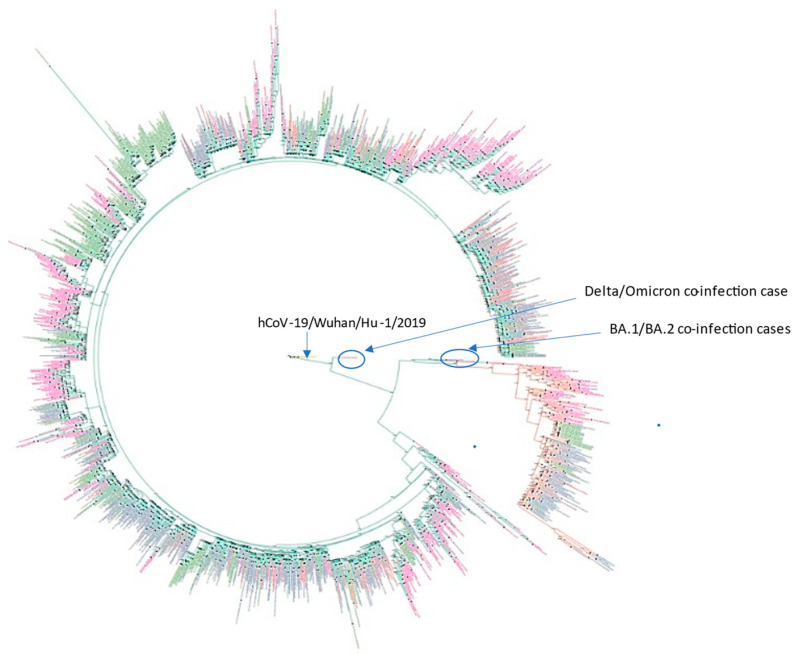
Phylogenetic tree built using the maximum likelihood method, including the Russian Moscow and GISAID datasets, and a global subsampled GISAID dataset. BA.1 cluster branches are marked with green, BA.2 branches with orange. The identifiers of the Russian strains from the GISAID dataset are marked with green; Russian Moscow, pink, Great Britain, blue; and USA, orange. More detailed information is available in the Appendix A.

**Figure 3 pathogens-11-01461-f003:**
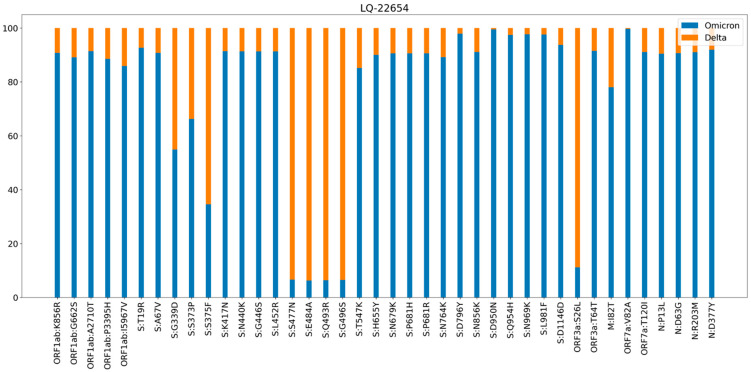
Proportion of reads across the genome corresponding to Delta (orange) and Omicron (blue) variants identified in the sample LQ-22654. Each bar is a position in the genome, marked by a corresponding gene and amino acid substitution.

**Figure 4 pathogens-11-01461-f004:**
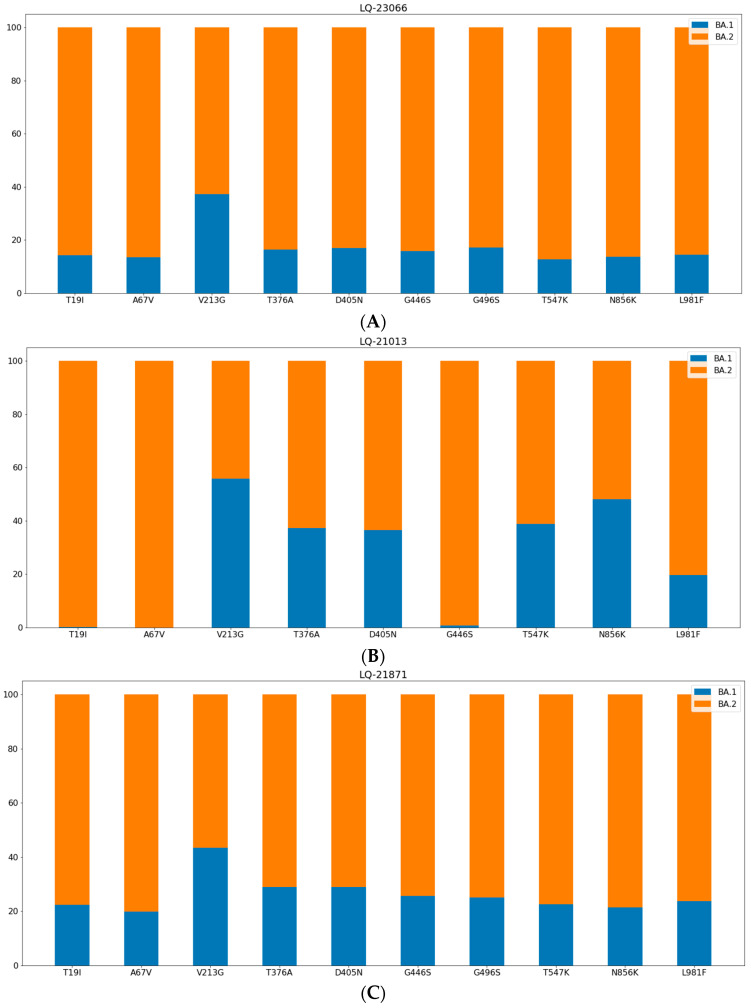
Proportion of reads in S gene corresponding BA.1 (blue) and BA.2 (orange) variants identified in t samples LQ-23066 (**A**), LQ-21013 (**B**), and LQ-21871 (**C**).

**Figure 5 pathogens-11-01461-f005:**
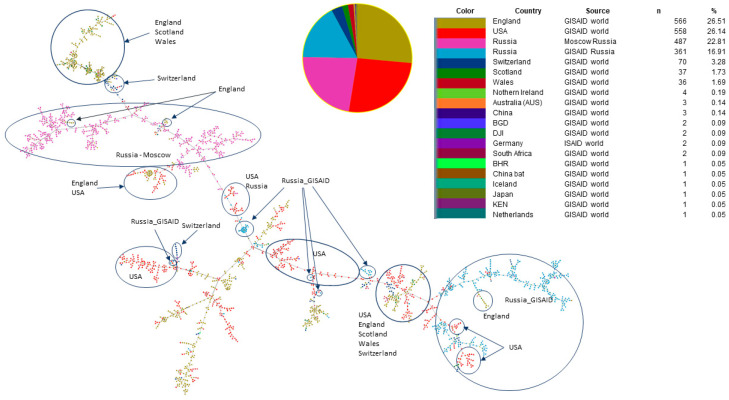
The structure of genomic transmission of BA.1 strains, including genomes of the Russian isolates from Moscow, Russian isolate from GISAID, and closely related genomes from the global dataset identified with UShER.

**Figure 6 pathogens-11-01461-f006:**
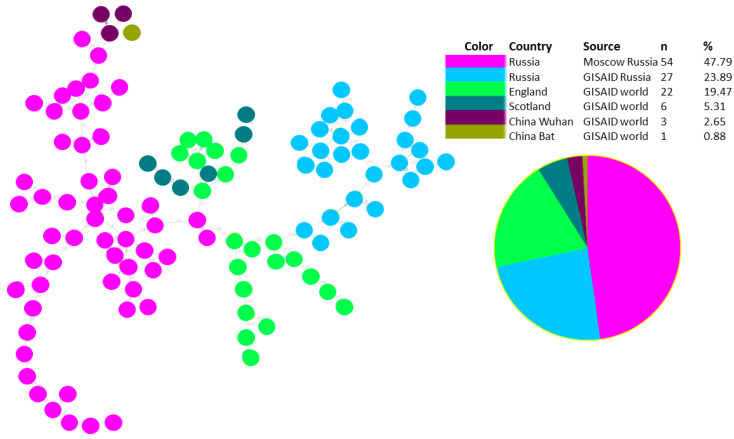
The structure of genomic transmission of BA.2 strains, including genomes of the Russian isolates from Moscow, Russian isolates from GISAID, and closely related genomes from the global dataset identified with UShER.

**Figure 7 pathogens-11-01461-f007:**
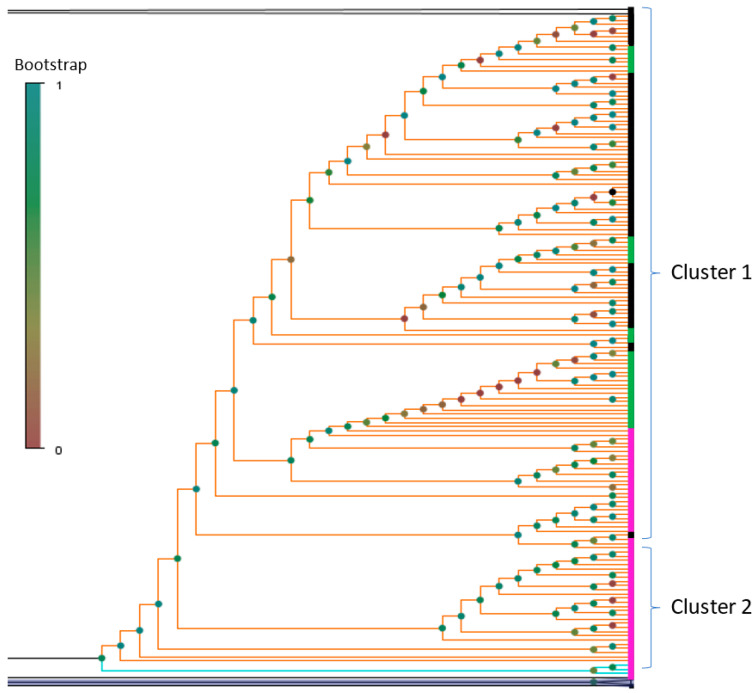
BA.2 cluster showing the relatedness of the Russian SARS-CoV-2 Omicron strains with the most closely related isolates from the GISAID dataset. BA2 cluster is marked with orange branches, strains from coinfected patients are associated with light blue branches. Russian Moscow strains are marked with the pink color font, Russian GISAID Omicron strains are marked with a green color, global isolates from other countries are marked with a black. Bootstrap for each inner node is marked with a circle coloured according to the values.

**Figure 8 pathogens-11-01461-f008:**
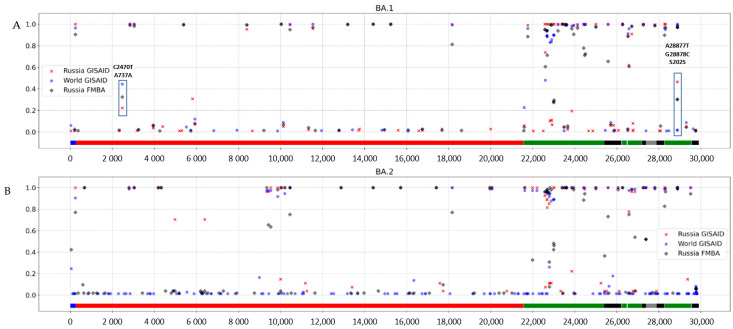
Frequency of nucleotide substitutions across the SARS-CoV-2 genome identified in different datasets of BA.1 and BA.2 lineages. (**A**) Omicron BA.1. (**B**) Omicron BA.2.

**Figure 9 pathogens-11-01461-f009:**
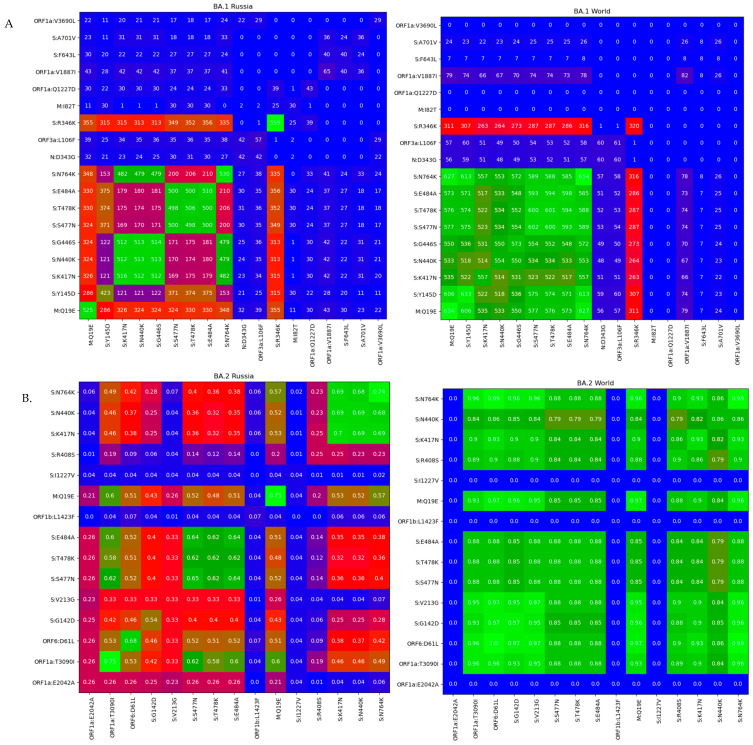
Heatmap visualization of coexisting amino acid substitutions identified in different lineages. The frequency of substitution coexistence is described in two datasets of SARS-CoV-2 strais: global and Russian. (**A**) Amino acid substitutions in BA.1 lineage in the Russian and global population. (**B**) Amino acid substitutions in BA.2. lineage in the Russian and global population.

**Table 1 pathogens-11-01461-t001:** SARS-CoV-2 genomes used in this study for the comparative analysis.

Source	Number of Samples
FMBA	542
GISAID global	756
GISAID Russian	397
UShER	1702
Wuhan genomes	4
Bat coronavirus RaTG13	1

**Table 2 pathogens-11-01461-t002:** Phylogenetic groups identified by phylogenetic analysis.

Group of Strains	Number of BA.1 Strains	Number of BA.2 Strains	Number of Non-Clustered Strains
Russian Moscow	483	55	4
Russian GISAID	370	27	0
Global sample GISAID	683	73	0
Total	1536	155	4

**Table 3 pathogens-11-01461-t003:** Delta-specific substitutions identified in Omicron strains in Russian and global subsampled GISAID dataset.

Substitution	Number of Genomes	Pangolin Classification
M:I82T	18	BA.1, BA.1.1., BA.1.1.13, BA.1.1.15, BA.2
S:L452R	12	BA.1, BA.1.1, BA.1.17
S:L452R, M:I82T	12	BA.1, BA.1.1, BA.1.17,
N:R203M	1	BA.1.1
S:L452R, M:I82T, ORF1a:P2046L	1	BA.1.1

**Table 4 pathogens-11-01461-t004:** Neutralizing epitope analysis based on BA.1 and BA.2 SARS-CoV-2 lineages genome sequences.

Type of Epitope	BA.1 Mutations	BA.2 Mutations	Neutralizing Epitopes in BA.1 and BA.2	Total
Discontinuous Epitopes Number	228	196	82	311
Linear Epitopes Number	3	3	5	8

## Data Availability

Fasta files were submitted to GISAID database, metadata and information about uploaded genomes are presented in Appendix A.

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
