# Peer review of "Genomic Variability of SARS-CoV-2 Omicron Variant Circulating in the Russian Federation during Early December 2021 and Late January 2022"

_pathogens, 2022, doi:10.3390/pathogens11121461_

Round 1

Reviewer 1 Report (Previous Reviewer 1)

The MS has been approved, thanks to authors that they`ve cleared my major concerns in the text.

There are minor issues which they can be addressed during the proof stage before the publication; 

The figure legend of fig4 was repeated, please remove the first one, the figures in the Fig 4 must be labelled as a, b and c as cited in the text. Please cite Figures in the Figs 5 and 6 as A and B as well as in the legends.

Please cite A and B labels in the legend of fig 8. 

The table 3 has not been cited in the text (it might be cited in the text numbers between 442 and 454). 

Author Response

Comment: The figure legend of fig4 was repeated, please remove the first one. the figures in the Fig 4 must be labelled as a, b and c as cited in the text. Please cite Figures in the Figs 5 and 6 as A and B as well as in the legends.

Response: The text is changed: we changed mention of the fig.4 in the text and labels ABC are added, lines 259-261

Comment: Please cite A and B labels in the legend of fig 8. 

Response: Changed, citations are highlighted with red font, lines 413-414, 424, 427

Comment: The table 3 has not been cited in the text (it might be cited in the text numbers between 442 and 454)”. 

Changed, citations are highlighted with red font, line 458 (now its table 4)

Reviewer 2 Report (Previous Reviewer 2)

the paper is undoubtedly improved. Results and discussion are better structured and documented than the previous version. I would recommend making a summary table showing the number of strains analyzed in USher, GISAID, FMBA to make interpretation of the results easier. Finally, I recommend a careful re-reading of the manuscript because there are still some writing errors.

For example:

Line 53: August 202(3)= August 202? (3)

Line 56: NSPS=NSPs

Line 68: idnentify= identify

Line 131; hCoV-19/Wuhan/Hu-1/2019 и hCoV-19/Wuhan/HB- 131 WH5-222/2020 = hCoV-19/Wuhan/Hu-1/2019 and hCoV-19/Wuhan/HB- 131 WH5-222/2020

Line 138 the citation of USher is missing

and others

Author Response

Comment: I would recommend making a summary table showing the number of strains analyzed in USher, GISAID, FMBA to make interpretation of the results easier.

Response: The table was introduced in the section Methods and materials. Lines 159-163

Comment: Finally, I recommend a careful re-reading of the manuscript because there are still some writing errors ...

 Response: thank you for notes, we have changed the text and found several additional typos

This manuscript is a resubmission of an earlier submission. The following is a list of the peer review reports and author responses from that submission.

Round 1

Reviewer 1 Report

The submission contains the information of the genomic variabilities of SARS-CoV2 variants spread in Russia. The information provided in this study might have some merit but I have some several and serious concerns about the study; 

First and most importantly informed consent, an Institutional review statement and approval of the study must be obtained to ensure ethics in research. 

The introduction shouldn`t be sub-sectioned. It should be whole and must contain only relevant information about the study and topic.

The result section is difficult to follow and some figures or tables that are cited in the text either don`t contain the relevant information cited for or are weak to understand. Such as Fig 1 is cited for Maximum Likelihood estimation based on WGS data but Fig 1 is actually showing the collected Omicron samples in Russia with numbers on time scale. Same as for tables such as Table 3 has been cited for identified mutations with freq. but table 3 is presenting the result of neutralizing epitope numbers. The place of figures are not correct such as Figure 4 has given after figure 9. Some figures are difficult to read such as Figure 7 and 8. Some figures are containing compressed data such as Fig 2 and Fig 6 and 8.  Some text in the results doesn`t cite any figures or table such as lines between 6 and 23; 100 and 109; and there are many that says 

“The plot shows..’ ‘Genome sequence data shows that’ ‘reads to the reference showed that’ ‘PHYLOViZ tool shows’ ‘UShER dataset showed that’ without citation of relevant figure or table.  Supplementary data depository containing two different identical figure names such as S Fig 1 and 2 as a pdf and Sipplementary figures docx file. I don`t understand why authors choose the present their results in such way that hard to understand. 

I even did not go through reviewing the discussion due to not fully able to see the results and understand of the presented data in this submitted draft.

Author Response

We would like to thank the reviewer for the time spent for the analysis and valuable comments. These comments have now been taken into consideration. Below we have provided separate responses for each point of the comments.

Comment: First and most importantly informed consent, an Institutional review statement and approval of the study must be obtained to ensure ethics in research. 

Response: According to the reviewer request all personal information was removed from the text. The rest data don’t open any with personal information and don’t require inform consent.

Comment: The introduction shouldn`t be sub-sectioned. It should be whole and must contain only relevant information about the study and topic.

Response: At the request of the reviewer Introduction section was merged, the text of the section contain information that is reasonable in our opinion

Comment: The result section is difficult to follow and some figures or tables that are cited in the text either don`t contain the relevant information cited for or are weak to understand. Such as Fig 1 is cited for Maximum Likelihood estimation based on WGS data but Fig 1 is actually showing the collected Omicron samples in Russia with numbers on time scale.

Response: Agree. We have changed the citation of the figure.

Comment: Same as for tables such as Table 3 has been cited for identified mutations with freq. but table 3 is presenting the result of neutralizing epitope numbers.

Response: Agree. We have changed the citation of the table.

Comment: The place of figures are not correct such as Figure 4 has given after figure 9.

Response: Figure 8 after figure 9 Agree, we have changed number of the figure and mistake in the citation of figure 9.

Comment: Some figures are difficult to read such as Figure 7 and 8.

Response: Agree. Figure 7 was changed. As soon as Figure 8 showed specific fragments of the large phylogenetic tree shown in supplementary figure 1, we left only description of specific group of samples and gave a reference to supplementary material.

Comment: Some figures are containing compressed data such as Fig 2 and Fig 6 and 8. 

Response: Agree. Fig 8 was removed. Fig.2 shows a phylogenetic tree constructed using very large dataset, and fig.6 is an output from Phyloviz software. Both figures 2 and 6 which does not allow to reflect it completely all details. however, the basic information is reflected and might be clear enough to illustrate the text.

 Comment: ome text in the results doesn`t cite any figures or table such as lines between 6 and 23; 100 and 109; and there are many that says 

Response: Agree. Figure and table numbers and references in the text were corrected. We added citation of figure 1 and figure 7 in the text to make it clearer, however we will add more citations of figures or tables if necessary.

Reviewer 2 Report

The article is interesting for the analytical approach used to study the genetic variability of the Omicron variant in the Russian Federation. Unfortunately, it is limited to a short period of time which does not allow to analyze the evolution of all omicron'sublineages. It would be interesting to extend the study beyond January 2022 so as to also analyze the spread of the other omicron sub-lineages.

I suggest the authors to add in the title that the study analyzes the omicron variants circulating in the Russian Federation during early December 2021 and late January 2022.

In the introduction, the authors write "currently, the omicron lineages contains two sublineages: BA.1 and BA.2". In the light of current knowledge, we know that there are more circulating lineages (such as, BA.4, BA.5, BA.3). I suggest to replace "currently" with "until late January 2022".I recommend rewriting the introduction by expanding it and possibly updating it.

Regarding co-infections, were the samples re-sequenced in order to have further confirmation? What results did the PCR assays give on these samples?

There are some mispelling such as:

-in "Data prepartion and Phylogenetic analyis" in line 7 there is a letter in Cyrillic

-in table 2: PANLOLIN instead of PANGOLIN

Author Response

We would like to thank the reviewer for the time spent for the analysis and valuable comments. These comments have now been taken into consideration. Below we have provided separate responses for each point of the comments.

Comment: I suggest the authors to add in the title that the study analyzes the omicron variants circulating in the Russian Federation during early December 2021 and late January 2022.

Response: Agree, the title has been changed

Comment: In the introduction, the authors write "currently, the omicron lineages contains two sublineages: BA.1 and BA.2". In the light of current knowledge, we know that there are more circulating lineages (such as, BA.4, BA.5, BA.3). I suggest to replace "currently" with "until late January 2022".

Response: Agree, we have changed the text in accordance with recommendation

Comment: I recommend rewriting the introduction by expanding it and possibly updating it.

Response: Agree. A new fragment of the text is added into the Introduction giving updated information on Omicron subvariants spread after January 2022

Comment: Regarding co-infections, were the samples re-sequenced in order to have further confirmation? What results did the PCR assays give on these samples?

Response: During our WGS analysis we identified two suspicious samples LQ-22654 and LQ-23066. Both had mutations associated with omicron in RdRp region P681H and N679K, but also mutations A67V and 67-70 del in spike and LQ-22654 carried L452R. Although we identified Delta-specific mutations in omicron strains these isolates were outliers in phylogenetic analysis. These two samples were re-sequenced and also for this sequencing run several strains with unusual combination of mutations in PCR were added, thus we identified two other cases of co-infections.

Information about mutations identified in PCR test was added to the Results section of the manuscript.

Comment^ There are some mispelling such as:

-in "Data prepartion and Phylogenetic analyis" in line 7 there is a letter in Cyrillic

-in table 2: PANLOLIN instead of PANGOLIN

Response: Agree, the text is corrected

Reviewer 3 Report

Dear authors,

the manuscript is focused on a exhaustive characterization of the variability of the Omicron clade by analysing its different sublineages in the Russian Federation during early December 2021 and late January 2022. The majority of viral strains belong to the BA.1 lineage and its numerous sub-lineages due to its earlier introduction in Russia. The paper is well written and the results are quite interesting but there are some points that deserve a revision.

Some brief comments:

1) page 1 title, page 2 line 48-56, page 4 line 163, page 5 line 228, page 6 line 250-254, and so on; why is the text in red color? Please change it.

2) the page numbering is wrong; please change it.

3) page 3 line 132: please add to median coverage min and max values or IQR.

4) page 3 187-188 line: the sentence “thus, in the 1699 tree there about 120 countries, while in GISAID only 17” maybe is wrong. According to the previous sentences the 1699 set seems to be the GISAID one.

5) page 5 line 213: the criteria for selection of BA.1 and BA.2 samples seems to be wrong, because variant BA.1 usually harbour the mutation A67V and you mention that the absence of this mutation led your sample selection.

5) page 6 line 251-252: regarding LQ-22654 sample, could you please give some information about the lineage? It sounds as BA.4/BA.5 due to its mutational pattern. Did you tried to set up a phylogenetic tree using some BA.4/BA.5 sequences from GISAID?

6) about the risk of laboratory contamination, due to the strong increase of the number of swabs to test in that period characterized by the escalation of Omicron variant, are you sure about it since you assert that “no other contamination/co-infection cases were identified”? Did you repeat the tests?

7) figure 3: it isn’t clear how you attributed to Delta or Omicron variant the proportion of reads for each mutation. Therefore, it’s strange that the majority of reads of L452R (signature mutation of Delta variant) belongs to Omicron and that the majority of reads of E484A (signature mutation of Omicron variant) belongs to Delta. In addition how can you explain the same proportion of reads for two different mutation at codon 681 (P681H signature of Omicron and P681R signature of Delta variant)?

8) the same doubt for the analysis of BA.1 and BA.2 (figure 4).

Round 2

Reviewer 1 Report

The majority concerns of mine have been met. However, there is some major issues remaining to clarify. 

As I`ve mentioned before Institutional Review Board Statement and Informed Consent Statement must be obtained and provided. Any study involved human or animal including samples taken from them subject to ethical regulations.  

As I`ve addressed before, the introduction section must be improved. Lines from 57 to 89 is more relevant to discussion section. I recommend keep the section about the virus (67 to 72) and complete the rest according to known and unknown knowledge in the field that you aimed to fill that gap. 

A conclusion section need to be included after Discussion.